# Characteristics and Health Risk Assessment of Heavy Metal Pollution in Haikou Bay and Adjacent Seas

**DOI:** 10.3390/ijerph19137896

**Published:** 2022-06-27

**Authors:** Dezhong Wang, Honghai Zhang, Wenzhuo Zhu, Xiaoling Zhang, Qiao Yang, Mei Liu, Qingguo Chen

**Affiliations:** 1Department of Marine Chemistry, College of Marine Science and Technology, Zhejiang Ocean University, Zhoushan 316022, China; wangdezhong@zjou.edu.cn (D.W.); zhangxiaoling@zjou.edu.cn (X.Z.); 2Key Laboratory of Marine Chemistry Theory and Technology, Ocean University of China, Ministry of Education, Qingdao 266100, China; honghaizhang@ouc.edu.cn; 3ABI Group, Zhejiang Ocean University, Zhoushan 316022, China; qiaoyang1979@whu.edu.cn; 4Zhejiang Provincial Key Laboratory of Petrochemical Pollution Control, Zhejiang Ocean University, Zhoushan 316022, China; liumei@zjou.edu.cn (M.L.); qgchen@zjou.edu.cn (Q.C.)

**Keywords:** heavy metals, correlation analysis, seawater, surface sediment, fish, Haikou Bay

## Abstract

Heavy metal contamination in coastal waters may pose a serious threat to aquatic products and human health. This study aimed to gain a better understanding of the pollution-induced by heavy metals in Haikou Bay and adjacent seas and assessed the potential ecological risk. The spatial distributions of heavy metals including Cu, Pb, Zn, Cd, Cr, Hg, and As were analyzed in the surface and bottom water, surface sediment, and five species of fish collected from Haikou Bay and adjacent seas. For seawater, the results showed that the horizontal distribution of the seven heavy metal elements in the study area had no uniform pattern due to the influence of complex factors, such as land-based runoff, port shipping, and ocean current movement. In contrast, the vertical distribution of these heavy metal elements, except for Zn and Cd, showed high concentrations in the surface water and low concentrations in the bottom water. Due to the symbiotic relationship between Zn and Cd, the distributions of these two elements were similar in the study areas. Different from the complex distribution of heavy metals in water, the highest concentrations of these elements in surface sediment all occurred at station 11 except for Pb. Our study revealed that organic carbon and sulfide are important factors affecting the heavy metal concentrations in the surface sediments. Heavy metals in waters and surface sediment were lower than the quality standard of class I according to the China National Standard for Seawater Quality and the sediment quality, except for Zn in water, suggesting that the seawater and surface sediment in Haikou Bay and adjacent seas has not been polluted by heavy metals. Additionally, the heavy metal As was the main element affecting the quality of fish in this study area, and attention should be paid in the future. The target hazard quotient (THQ) values of seven heavy metal elements in fish were all lower than 1.0, indicating that eating fish in this area will not pose a risk to human health. These results provide valuable information for further understanding the status of heavy metal pollution in Haikou Bay and adjacent seas and the development of targeted conversation measures for the environment and fish consumers.

## 1. Introduction

In the past 50 years, with high-intensity industrial and agricultural production, urban construction, and other similar activities in the coastal zone, a large number of heavy metals have entered seawaters and attached to the suspended substances which then settle on the floor of the bay, eventually mixing with the water itself [1,2,3]. At the same time, the heavy metals in the sediment also can be re-released into the water as the hydrodynamics or redox of the water changes. This process led to secondary pollution, and heavy metals were released by sediment [4,5,6]. Heavy metals have become a class of contaminants that cannot be ignored in the marine environment due to their persistent environmental impact, biogeochemical recyclability, and bioaccumulation [7,8,9,10]. With the continuous improvement of living standards, people have paid more and more attention to food safety. However, as the marine environment was polluted by heavy metals, the originally high-quality marine foods, such as fish, become harmful substances that endanger the health of people who eat them [11]. Many scholars study and evaluate marine environmental quality through marine organisms [8,12,13,14,15]. At present, there are few reports on comprehensive studies on seawater, surface sediment, and fish in Haikzou Bay [16,17].

Hainan is the second largest island in China and has abundant marine resources. Haikou Bay and its adjacent waters are located in the northern part of Hainan Island and the south of the middle section of the Qiongzhou Strait. It is a channel area between the South China Sea fishery and the Beibu Gulf fishery among the four traditional Chinese fisheries. It is rich in fishery resources. Haikou Bay is about 14 km long from east to west, about 6 km wide from north to south, with a coastline about 20.5 km in length and a total area of 42 km^2^ [18]. The study area is dominated by wind and waves throughout the year. The normal wave direction is in the NEE direction, and the sub-normal wave direction is in the NE direction. The tide is irregular diurnal tide, and the current is also irregular diurnal tide. The largest rising tide is generally eastward, and the falling tide is westward, showing a reciprocating flow. With the economic and social development of Haikou City, especially since the development and construction of Hainan International Tourism Island, large-scale coastal tourism and marine industry development have led to population aggregation and high resource consumption, making the coastal waters of Haikou Bay increasingly polluted [19]. Existing studies have paid more attention to the pollution characteristics of seawater or surface sediments [17,20,21,22,23,24], while comprehensive studies on the distribution characteristics, pollution sources and influencing factors of heavy metals in seawater, surface sediments, and fish are relatively lacking. Therefore, in this study, the concentrations of seven heavy metal elements (Cu, Pb, Zn, Cd, Cr, Hg, and As) in the surface and bottom water, surface sediments, and five species of fish in Haikou Bay and adjacent bodies of water were analyzed. To describe the distribution characteristics of the seven heavy metal concentrations and reveal the pollution status of heavy metals in Haikou Bay, this study researched the possible sources and influencing factors and comprehensively evaluates the health risks that are significant for protecting the health of fish consumers in the area. These results can effectively indicate the degree of heavy metal pollution in waters, sediments, and fishes in Haikou Bay, and provide a theoretical basis for preventing heavy metal pollution. The purpose of this study is to (1) discuss the sources of heavy metals in the surface and bottom water, sediment, and fish in Haikou Bay and adjacent seas; (2) explore the factors that affected the concentration of heavy metals in the study area; (3) assess the risk of heavy metals pollution in this study area.

## 2. Materials and Methods

### 2.1. Sampling and Sample Pretreatment

In April 2016, a total of 16 sampling points for a water quality survey were set up in Haikou Bay and the adjacent sea areas (110°06′41.34″~110°24′51.90″ E, 20°02′42.30″~20°12′04.80″ N) with seven sampling points for sediment surveys; in May of the same year, five sampling points for a fish quality survey were set up (Figure 1). The water samples from both 2 m below the surface and 2 m above the seafloor were collected using a Niskin sampler (General Oceanics, Miami, FL, USA). The water samples for metal determination were filtered immediately through a 0.45-μm pore glass fiber filter (GF/F, Whatman) and acidified with nitric acid to pH 1.5–2.0 for preparation for analysis. The filters and the water samples were stored in a −20 °C refrigerator. Suspended particulate material (SS) samples were filtered through pre-weighed Whatman GF/F fiber filters (25 mm). The SS samples were dried and weighed to determine the amount in mg/L of sample. The seawater for dissolved oxygen (DO) analysis was collected with a tube reaching the bottom of the bottle until the water overflowed. DO was determined using the Winkler titration method according to Gao and Song (2008) [25] just after sampling. The seawater temperature and salinity were measured in situ with a multiparameter sensor YSI6600, and pH values were determined with a pH meter. Active phosphorus (AP) was measured using the approach according to Valderrama (1981).

Sediment samples at each sampling point were collected with a grab sampler (China Kelan Ocean), and the samples were placed in a freeze dryer (Shandas-48, China) at −75 °C and 5 Pa to dry. The samples were dried by hand-made agate Mortar (China LC) that was ground to powder, put through a 106-μm nylon sieve, and stored until it was ready for digestion. The determination of organic carbon (OC) in sediments was by potassium dichromate oxidation-ferrous sulphate titrimetry. The contents of sulfide and petroleum hydrocarbon (TPH) in sediment were measured by spectrophotometry.

The scientific research vessel captured five local representative fish samples at five trawl stations and analyzed six fish samples in total. The collected samples were placed in a polyethylene bag with the air pressed out, and the bag mouth was knotted. The bag and sample label were put into another polyethylene bag, sealed, frozen, and brought back to the laboratory for analysis. The muscle tissue of the samples was dried, ground, put through a sieve, and stored until it was ready for digestion in the same manner as the pellet samples.

### 2.2. Determination of Metals

When measuring Cu, Cd, Pb, Zn, and Cr, the sample was mixed with amine dithiocarbonate (APDC) and diethyldithiocarbamate. The ethylamine (DDDC) mixed solvent was chelated and then extracted with CHCl_3_, and the content of the five elements in the extract was determined by the atomic absorption method (ICP-MS, VG elemental PQII). To determine the level of Hg and As in the water sample, 25 mL of the water sample was added to a glass tube with 2 mL of a potassium borohydride reducing agent, and an atomic fluorescence spectrometer (AFS XGY-1011A, China) was used under the action of the carrier gas N_2;_ the mercury vapor (for mercury)/hydrogen (for arsenic) arsenide formed was determined by an atomic fluorescence spectrometer (AFS XGY-1011A, China).

The determination of heavy metals in sediments was as follows. The Teflon digestion vessel containing 0.2 g of dry sediment samples was added to 6 mL of nitric acid, 2 mL of perchloric acid (China, chemical reduction method), and 1 mL of hydrochloric acid (China, chemical reduction method) and left for 24 h for the process of simplification. The digestion vessel was then transferred to a microwave digestion apparatus (Anton PAAR, Austria), and the digestion was performed according to a four-step procedure (180 °C 800 watts for 10 min, 190 °C 900 watts for 10 min, and finally 100 °C 400 watts for 10 min). After digestion, the vessel was heated to 80 °C until the remaining acid reached a level of approximately 1 mL. The remaining solution was transferred to a polyethylene tube (LABSBUY, China) and diluted to 25 mL with Milli-Q water (MILLIPORE, USA). The diluted solution was filtered using a syringe filter (SFPTFE013022SL-13, pore size = 45 mm, ASONE, Tokyo, Japan) and was stored in a 25 mL polyethylene tube. The presence of copper, lead, zinc, cadmium, and chromium in the digestion solution was determined by the atomic absorption spectrometry, and the presence of mercury and arsenic was determined by the atomic fluorescence spectrometry.

The fish samples were digested by the HNO_3_-HClO_4_ wet method. Fish samples weighing 0.5 g were put into a digestion tube. 10 mL of nitric acid and 1 mL of perchloric acid were added to the digestion tube to digest for five hours at 130 ~ 180 °C until the solid sample was completely digested into a liquid; itwas then taken out to cool at room temperature. A small amount of nitric acid of up to 50 mL was added, shaken well, and put into the refrigerator for later use. A reagent blank test was completed at the same time. A total of six samples of five types of common local economic fish were collected, and individuals of similar size of each species were selected for metal analysis. The muscular tissues of the samples were dried, ground, sieved, and stored until digestion in the same way as the sediment samples.

### 2.3. Analytical Methods

According to the previous relevant research conclusions [5,26,27,28,29,30,31,32], this study used the weighted modified Nemerov pollution comprehensive index to evaluate the seawater quality categories, single-factor pollution index, comprehensive pollution index and potential ecological risk coefficient. The pollution status of heavy metals in the sediment was evaluated; similarly, the single-factor pollution index evaluation method was used to evaluate the pollution status of heavy metals in fish. The hazard quotient was reused to evaluate the human consumption of fish in this sea area for possible health risks. The carcinogenic risks (CR) of the heavy metals in the fish samples were used to assess the health risks of daily consumption of the fish. The calculation method is shown in the following Table 1.

### 2.4. Data Processing

The Ocean Data View software was used to draw the schematic diagram of the monitoring station. Excel 2013 and IBM SPSS 26.0 software were used for data processing and analysis, and Origin 2018 software (Origin Lab, Northampton, MA, USA) was used to draw the data.

## 3. Results and Discussion

### 3.1. Heavy Metals in Water

The concentrations of seven heavy metals in waters in Haikou Bay and the adjacent areas were shown in Table 2 and Figure 2. The concentration of Zn at some sampling points (1, 2, 8, 9, 11, 13, 15) exceeded the first-class standard of China’s seawater quality ($P_{s}^{’}\;$= 0.754, $P_{b}^{’\;}$= 0.752, Table 2), and accounted for 18.75% and 25% of the total concentrations of heavy metals in surface and bottom water, respectively. The highest concentration of Zn was 21.3 µg/L, which was slightly higher than the first-class standard (20 µg/L). The average concentration of heavy metals in the surface and bottom water was in the following order: Zn >As > Cu > Cr> Pb> Cd> Hg. Therefore, most heavy metals have reached the national first-class standard for seawater quality in this study area.

The vertical distribution of these heavy metal elements, except for Zn and Cd, showed high concentrations in the surface water and low concentrations in the bottom water. Combined with the background trend of seawater salinity gradually increasing from surface to bottom in this study area, these findings were consistent with Wang et al. (2007) [35], who indicated that the concentration of heavy metals decreased with the increase of seawater salinity.

There were obvious differences in the spatial distribution characteristics of the seven heavy metals in surface and bottom seawater (Figure 2). The overall trend of the Cu was highest in the east and lowest in the west of Haikou Bay. The high-value of Cu in surface water was located in the eastern part of the study area and showed a decreasing trend from east to west. This may be influenced by the spring current, which mainly flows from east to west. The high-value of Cu in bottom water was located in the local area of the near-shore estuary of Haikou Bay, which may be related to the large amount of rock weathering products carried by the Nandu River and its tributary Henggou River, as well as the contribution of the near-shore sewage outlet and drainage ditch. Moreover, the ebb and flow tide at the estuary weaken the flow velocity of the runoff into the sea. The above factors influenced the distribution characteristics of high concentrations of bottom Cu in the peripheral area of the nearshore estuary [36,37]. The overall distribution of Pb concentration in seawater was not obvious, but the concentration of Pb in the bottom seawater showed a good trend of decreasing from nearshore to offshore, which was similar to the results of Wei et al. (2004) [38] who showed that when rivers carry pollutants into the bay, heavy metals will rapidly settle on the seafloor, or form complexes and remain in the water. Moreover, the change in seawater salinity was one of the non-negligible factors affecting the distribution of Pb concentration [39,40,41,42]. With the increase of salinity from the nearshore to offshore, the content of Cl- in seawater will gradually increase and can form a complex with Pb in seawater. The removal mechanisms such as flocculation, adsorption, and sedimentation of the complex will lead to the decrease of Pb content in seawater, and then lead to the concentration of Pb in the nearshore to the offshore decreasing. Some studies have also shown that Pb mainly comes from anthropogenic pollution sources, such as industrial wastewater discharge, shipping traffic, and sea runoff. In this study, the concentration of Pb in water was mainly affected by the Pb-containing exhaust gas emitted by the bulk cargo ships in Xiuying Port in the Bay or by the anti-corrosion paint containing Pb paint painted on the hull [43,44,45]. The overall distribution of Cd concentrations in seawater had no obvious distribution. The high-value of Cd in surface water was located in the western part of the study area and on the west side of the near-shore estuary of Haikou Bay, while the high-value of Cd in bottom water was distributed in the northern part of the study area. The north and south banks of the Qiongzhou Strait are covered with volcanic cones and lava covers, and the uneven distribution of Cd may be related to the settlement of atmospheric particulate matter formed by rock weathering, but the reason for this difference in distribution requires further investigation. The distribution of Zn high-value zones in seawater is similar to that of Cd, with a high concentration near the shore in the surface layer and a high concentration on the far shore in the bottom layer. This result may be due to the symbiotic relationship between cadmium ores and zinc ores. Generally, Cd exists in zinc ores in the form of CdS and CdCO_3_. In addition, the distribution of high-value zones at the surface and bottom layers of Hg was similar to that of Pb. The high-value of Hg in surface water was located in the middle of the study area, and the high-value of Hg in bottom water was located near the offshore artificial island in Haikou Bay. The surface high-value areas of As are distributed in the western part of the study area, and the bottom high-value areas are located near the Artificial Island, which is similar to the distribution of high-value areas of Pb and Hg. The concentrations of other heavy metal elements were lower where the surface layer of the Cr element has high values, indicating that there was a specific source of Cr in this area. The distribution of the overall high-value area of Cr was similar to that of the high-value area of Cu, indicating that the sources of these two elements may be the same or similar.

From Table 3 (Pearson correlation coefficient), the active phosphate (AP) in the surface water had a significant positive correlation with Cu, indicating that Cu is mainly affected by the complexation of organic matter. The active phosphate was also negatively correlated with dissolved oxygen (DO), indicating that oxygen consumption was important in the degradation process of organic pollutants in water. The active phosphate in water may be from the input of the Nandu River, which is the largest river in the region. It may also be related to the direct discharge into the sea from near-shore sewage outlets and spillways. There were significant negative correlations between Cu and As, and Pb and Cd, respectively, indicating that these two groups of elements have the same or similar sources of pollution. The heavy metals Zn, Hg, Cr, and other elements were not highly correlated, indicating that these three elements, in particular, may be different or come from more scattered sources of pollution. Similarly, the Pearson correlation coefficient in Table 4 showed that there was a significant positive correlation between Cu and Cr in the bottom water, indicating that the two elements had a similar source or the same geochemical process. DO had a very significant positive correlation with Pb and Hg, and DO also had a very significant negative correlation with active phosphate, which is consistent with the surface conditions. Pb had a significant positive correlation with Hg and both have a significant positive correlation with DO which may be due to the influence of the water-oxygen content of Pb and Hg in the water. The water-oxygen content affects the migration and transformation process of elemental Pb and Hg adsorption and release in the water environment.

### 3.2. Heavy Metals in Surface Sediments

The bay and offshore are the most important hosts of heavy metal pollutants [46,47]. Gulf sediments are the main reservoir of pollutants in the water and a potential source of pollution. Heavy metals are adsorbed by suspended solids in the water and eventually accumulate in sediments. Driven by a series of biogeochemical processes occurred at the water-sediment interface, heavy metals can re-enter the overlying water [48,49], and then pollute the water and fish.

The distributions of seven heavy metals in the surface sediments in Haikou Bay and adjacent sea areas are shown in Figure 3. The analysis results showed that the concentrations of heavy metals in the surface sediments at all stations meet the first-class standards for the National Marine Sediment Quality of China. The Cd gradually decreased from the nearshore to the far sea, while the spatial distribution characteristics of the other heavy metal elements were high in the east and low in the west. The high-value areas were distributed in the Nandu River Delta and the downstream area of the Henggou River estuary. The maximum concentrations appeared at the sampling point 11, which may be related to the fact that the sampling point 11 is located at the mouth of the Henggou River. The river’s aperture flow is supported by rising tides and waves, the runoff velocity is weakened, and the land-based source pollutants carried and migrated by the runoff are silted up here.

Both organic carbon and sulfide in the sediments had a significant positive correlation with the elements Cu, Zn, Hg, and Cr (Table 5), indicating that both organic carbon and sulfide are important factors that are affecting the behavior of heavy metals in the surface sediments in Haikou Bay and the adjacent sea areas. In this study, the highest concentrations of sulfide and the elemental Cu and Cd all appeared at station 11. At the same time, the lowest concentrations of sulfide and the elemental Cu all appeared at station 11. Previous studies have shown that heavy metals are easily removed from water when they react with organic carbon (mainly humic substances) and sulfides to form metal-organic complexes or metal sulfide precipitation through surface adsorption, cation exchange, and chelation reactions [40]. In this study, the concentration of organic carbon and sulfide reached its maximum value at the sampling point 11. At the same time, the concentrations of heavy metals Cu, Hg, and Cr in the surface water and the concentration of Zn in the bottom water at this point are lower than the average values in other sampling stations, which further indicates that the removal effect of organic carbon and sulfide on heavy metals may exist at the sampling station of 11. In addition, the elements Hg and Cr had a significant positive correlation with the elements Cu and Zn, and there was also a significant positive correlation between the elements Hg and Cr (*p* < 0.01, Table 5), indicating that the historical sources of these heavy metal elements in the surface sediments in this area are the same or very similar [30,50].

The single-factor pollution index method was used to evaluate the pollution degree of seven heavy metals in the sediments of the study area. The analysis results were shown in Table 6. From the statistical results of the heavy metal single-factor pollution index ($C_{f}^{i}$), the pollution index of each element was less than 1.0, indicating that the overall pollution level in this study area was low. However, the $C_{f}^{i}$ of elements Cu and Cr at the sampling point HS11 were as high as 0.95 and 0.97, respectively, which nearly breaches the safety value of 1.0 (Figure 4a). The contamination degree displayed by the mean value from low to high is Hg < Cd < Zn < Pb < As < Cr < Cu. Moreover, the comprehensive pollution index ($C_{d}$) evaluation results showed that the degree of pollution in the study area was at a low pollution level as a whole. The highest value of $C_{d}$ (4.20) appeared at the sampling station HS11 with a low degree of pollution, and the contribution rate of Cu and Cr was high, nearly 50%, while the proportion of other elements was relatively low.

The evaluation results of the potential ecological hazard index ($E_{r}^{i}$) of heavy metals in the surface sediments in Haikou Bay and adjacent sea areas are shown in Table 7. The statistical results of the $E_{r}^{i}$ of single heavy metal pollution and each element were less than 40, indicating that the potential ecological hazard of single heavy metal pollution in the study area was low. Except for the element Pb, the $E_{r}^{i}$ of other heavy metal elements station HS11 located near the Nandu River Delta were the highest, and the potential ecological risk was high. Furthermore, the comprehensive index of potential ecological hazards of heavy metal pollution (RI) showed that the potential ecological hazard of heavy metal pollution in surface sediment in this study was also at a low level (Table 7 and Table 8). At the same level, the contribution rate of Cd and Hg was high at nearly 65.9% and the proportion of other elements was relatively low. There was a big difference between the $E_{r}^{i}$ ranking and the $C_{f\;}^{i}$ ranking, which may be because the $E_{r}^{i}$ in considers the biological toxicity of heavy metals and can more accurately reflect the potential ecological hazards caused by heavy metals. The toxicity response coefficients of Hg and Cd were larger than other heavy metal elements and had higher potential ecological risk coefficients.

### 3.3. Heavy Metals in Fish

The concentrations of heavy metals in fish are shown in Table 9. The single-factor pollution index ($P_{i}$) was used to analyze the pollution level of heavy metals in fish in the study area (Figure 4b). The heavy metal As (1.16) in the yellow anchovy at station HS03 was greater than 1.0, indicating that the yellow anchovy at this station has been polluted by As. The single-factor heavy metal pollution index of the samples from other stations was, however, less than 1.0, indicating that the fish samples from other stations were not polluted by the detected heavy metals and were at the normal background value level. The pollution indices of the seven heavy metal elements from high to low were: As (0.58) > Hg (0.21) > Pb (0.15) > Zn (0.14) > Cd (0.07) > Cu (0.04) > Cr (0.01), indicating that the heavy metal As is the main element affecting the quality of fish in the study area. This should receive attention in the future. The comprehensive quality index ($P_{ij}$) can reflect the overall comprehensive quality of heavy metal pollution for the marine organisms in this area. From the calculation results, except for the value of the yellow anchovy $P_{ij}$ at Site 3, > 1, the fish in other stations $P_{ij}$ were all < 1 and were at low risk.

The THQ values of the seven heavy metals were all less than 1.0, indicating that eating wild fish in this area will not pose a risk to human health (Table 10). It is worth noting, however, that the THQ value of the heavy metal As reached 0.58, which should be researched in the future. The CR of Pb in fish at station 8, as well as Cd and As in fish at all stations, was greater than the permissible limit of 10^−6^ to 10^−4^. Cu, Zn, Cr and Hg have no cancer slope factor (CSF) and thus have no CR (Table 11). In order to effectively reduce the risk of cancer caused by ingesting heavy metals, consumers are advised to appropriately control the daily consumption of these seafood products and the number of meals per month [51]. The higher detection of As in fish requires further attention. The results of health risk assessment showed that the carcinogenic risk (CR) of heavy metal As in fish exceeded the acceptable range, and the maximum carcinogenic risk value was 1.09. It is suggested that consumers should reasonably control the consumption of these fish and the number of meals eaten. In addition, the paper cited the parameters of the United States, but there are great differences in the physical signs and living habits of the population of the United States and China, so there is some uncertainty in directly applying the relevant parameters of the United States to health risk assessment.

## 4. Conclusions

This field investigation revealed a higher concentration of seven heavy metals in sediment than in water in the Haikou Bay and adjacent seas. The higher concentrations of heavy metals were appeared at Nandu River and its tributary Henggou Estuary due to the deposition of land-based pollutants there. In particular, the concentration of heavy metals was high at the station 11 located in the Henggou River estuary, except for Pb. The heavy metal contents in fishes were lower than the legal thresholds. But station 3 should be given increased attention because its $P_{i}$ of As and $P_{ij}$ exceeded the risk thresholds. The degree of heavy metal pollution for fish and their habitats is important for preventing potential dangers that can be caused by heavy metal pollution and can also provide valuable information for the future management of the marine ecological environment in this study area.

## Figures and Tables

**Figure 1 ijerph-19-07896-f001:**
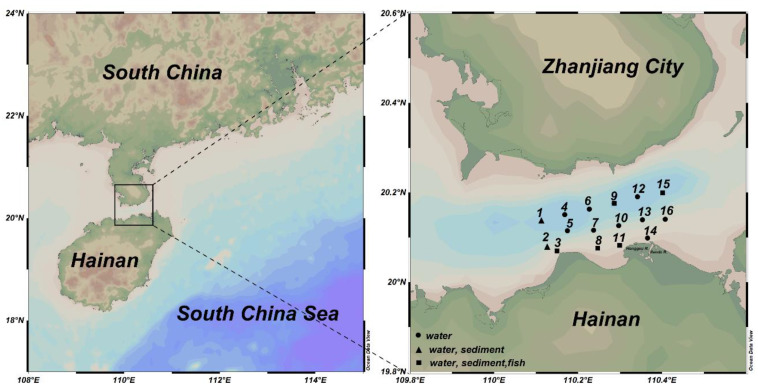
Schematic diagram of the study area and stations in Haikou Bay and adjacent seas.

**Figure 2 ijerph-19-07896-f002:**
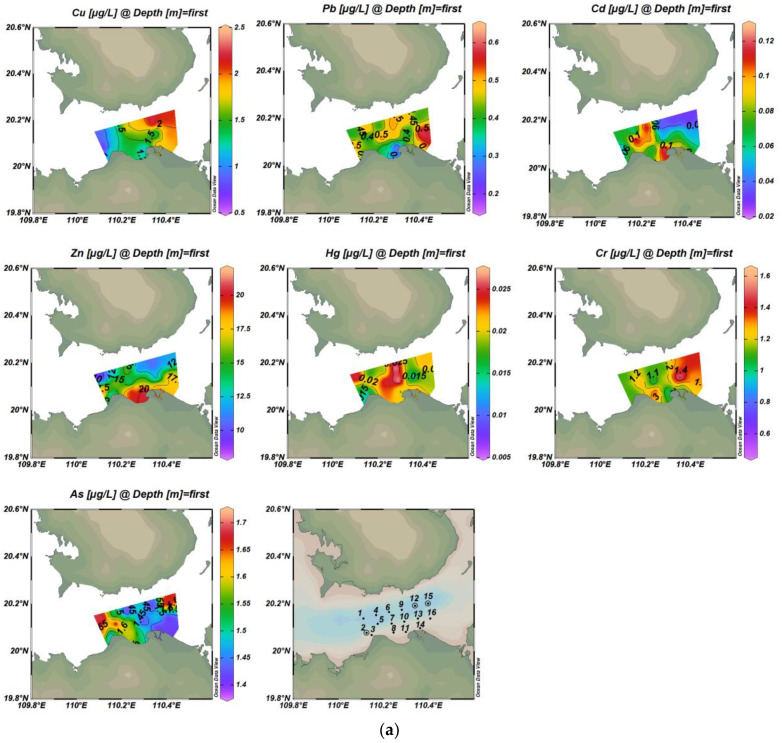
Spatial distribution of concentrations of seven trace metals in surface and bottom water bodies ((**a**) is the surface layer; (**b**) is the bottom layer).

**Figure 3 ijerph-19-07896-f003:**
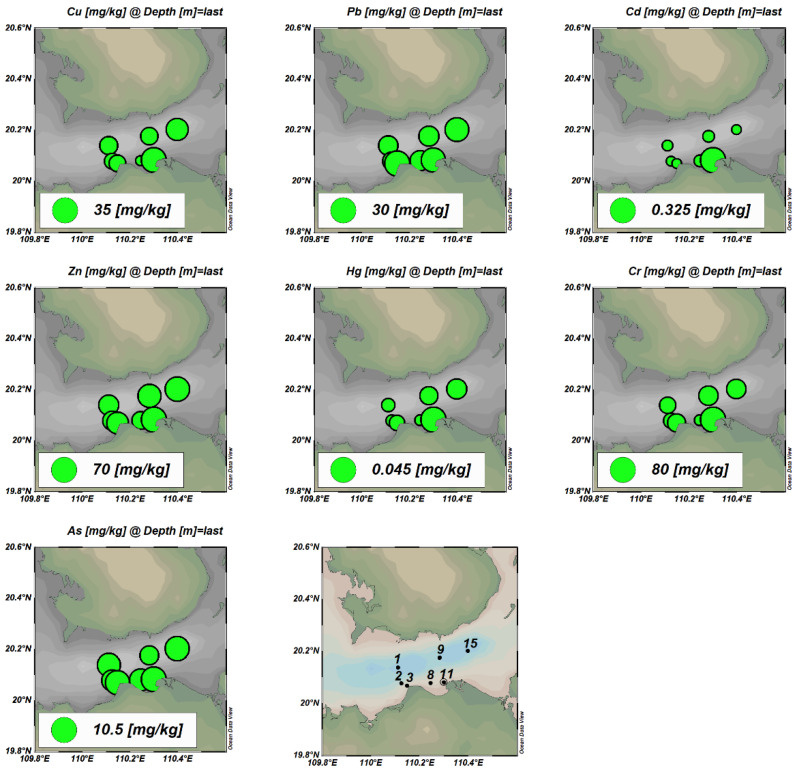
Spatial distribution of seven heavy metals in surface sediments.

**Figure 4 ijerph-19-07896-f004:**
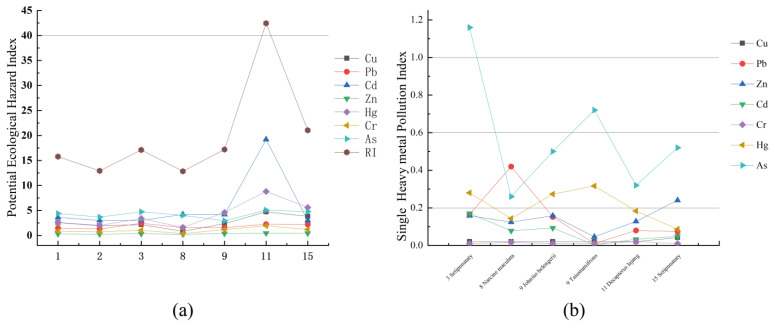
(**a**) potential ecological hazard index of sediment; (**b**) singe heavy metal pollution index of fish.

**Table 1 ijerph-19-07896-t001:** Parameters of pollution and health risk assessment.

Index	Description	Equation	References
Pollution assessment			
$P^{\prime }$	weighted to modify the Nemerom pollution comprehensive index ( $P^{\prime }$) If $P^{\prime }$ < 0.714, Class I water quality If 0.714 ≤ $P^{\prime }$ < 1.63, Class II water quality	$P^{\prime }$$=\sqrt{\frac{P_{ijmax}^{2}+P_{ijweighted\;average}^{2}}{2}}$ $P_{ijweighted\;average}$$=\frac{\sum_{i=1}^{n}\left(P_{ij\times W_{ij}}\right)}{n}$ $P_{ij}$$=C_{ij}$$/S_{ij}$$,\;W_{ij}$$=\frac{S_{\frac{jmax}{S_{ij}}}}{\sum_{i=1}^{n}S_{\frac{jmax}{S_{ij}}}}$ $P_{ijmax}$: monitoring point The maximum value of the pollution index $P_{ijweighted\;averrage}$: weighted average value of each pollution index at the monitoring point $P_{ij}$: the i-th pollution factor Type j pollution index $C_{ij}$: detected value of the i-th pollution factor (µg/L) $S_{ij}$: standard value of class J of the i-th pollution factor $W_{ij}$: the weight value of the i-th pollution factor $S_{jmax}$: the maximum value of the j-th standard value of all evaluation factors	Abdallah et al. (2014) [26] Luo et al. (2010) [29]
$E_{r}^{i}$	Potential ecological hazard index of heavy metal element I ($E_{r}^{i}$) $\mathrm{If}\;E_{r}^{i}$ is < 40, Potential ecological hazards is low 40 $\leq E_{r}^{i}\;$< 80, middle 80 $\leq E_{r}^{i}$< 160, higher	$E_{r}^{i}$$=T_{r}\times C_{f}^{i},C_{f}^{i}=\frac{C^{i}}{C_{n}^{i}}$$,\;C_{d}=\overset{n}{\underset{i=1}{\sum }}C_{f}^{i}$ $T_{r}$: toxicity coefficient of heavy metal element I $C_{d}$: comprehensive pollution index $C_{f}^{i}$: Single factor pollution index $C^{i}$: the measured content of the heavy metal element i in the sediment (×10^−6^) $C_{n}^{i}\;$: the evaluation reference value of the heavy metal element i (×10^−6^)	Luo et al. (2010) [29] Lei et al. (2013) [27]
RI	Comprehensive index of potential ecological hazards of multiple heavy metal pollution (RI) If RI is < 140, low 140 ≤ RI < 280, middle 280 ≤ RI < 560, higher	RI = $\overset{n}{\underset{i=1}{\sum }}T_{r}\times C_{f}^{i}$	Zhang et al. (2009) [30] Yahaya et al. (2021) [33]
$P_{i}$	Single factor pollution index of marine organisms ($P_{i}$) If $P_{i}$ is < 1, It indicates that the organism is not contaminated with this factor If $P_{i}$ is > 1, It indicates that organisms have been contaminated with this factor	$P_{i\;}$$=C_{i}$$/S_{ij}$ $C_{i}$: the concentration value of measured item I (mg/kg)	Gao et al. (2012) [28] Luo et al. (2010) [29]
$P_{ij}$	Comprehensive quality index of heavy metal pollution in marine organisms ($P_{ij}$) If $P_{ij}$ is > 3, Heavy pollution If 2 $<P_{ij}$≤ 3, Moderate pollution If 1 <$P_{ij}$≤ 2, Light pollution If $P_{ij}$≤ 1, pollution-free	$P_{ij\;}$$=\sqrt{\frac{{\left(maxP_{i}\right)}^{2}+{\left(aveP_{i}\right)}^{2}}{2}}$ $\;maxP_{i}$: the maximum individual quality index of the organism $\;aveP_{i}$: the average value of each quality index of the organism	Gao et al. (2012) [28] Luo et al. (2010) [29]
Health risk assessment			
THQ	Target hazard quotient (THQ) It assumes that the absorbed dose of trace metal for consumers is equal to the intake dose. When THQ is greater than or equal to 1, it indicates the likelihood of noncarcinogenic risk	THQ = EF × ED × FIR × c × 10^−3^/RFD × WAB × TA EF: the exposure frequency (365 days/year) ED: the exposure duration (70 years) FIR: the daily consumption of fish (36 g/day) c: the metal concentration in the fish (mg/kg) RFD: the oral reference dose (mg/(kg bw ·d)) WAB: the body weight (60 kg) TA: the average time of exposure to noncarcinogens (365 days year^−1^ × 70 years)	USEPA (2010) [34]
CR	Carcinogenic risks (CR) Acceptable lifetime cancer risk level ranges from 10^−^^4^ to 10^−^ ^6^	CR = CSF × DI, DI = EF × ED × FIR × CF × c/WAB × TA CSF: the oral carcinogenic slope factor (0.0085 mg/kg/d for Pb, 0.38 mg/kg/d for Cd, 0.015 mg/kg/d for i-As (we assumed that the inorganic As was 1% of the total As)) CF: conversion factor is 0.208, to convert the dry weight of fish to wet	USEPA (2010) [34]

**Table 2 ijerph-19-07896-t002:** Heavy metals concentrations in water from Haikou Bay and adjacent seas. Unit: µg/L.

		Cu	Pb	Cd	Zn	Hg	Cr	As
surface layer	Content range	0.6~2.3	0.23~0.61	0.03~0.13	8.5~21.3	0.012~0.028	1.0~1.6	1.4~1.7
	Average value	1.5	0.45	0.07	14.86	0.021	1.2	1.5
	Standard deviation	0.446	0.097	0.034	4.176	0.005	0.150	0.133
	Coefficient of variation	0.297	0.217	0.485	0.281	0.238	0.125	0.087
bottom layer	Content range	0.5~1.9	0.15~0.56	0.03~0.13	11.1~21.3	0.007~0.027	0.5~1.4	0.8~1.5
	Average value	0.9	0.24	0.07	16.2	0.017	0.9	1.0
	Standard deviation	0.386	0.115	0.032	3.728	0.005	0.274	0.158
	Coefficient of variation	0.413	0.484	0.441	0.231	0.301	0.322	0.154
LT		5	1	1	20	0.05	20	20
$P_{s}^{’}$		0.318	0.432	0.092	0.754	0.479	0.055	0.061
$P_{b}^{’}$		0.262	0.396	0.092	0.752	0.440	0.049	0.049

Note: the legislation thresholds (LT) mean the first-class standard of China’s seawater quality.

**Table 3 ijerph-19-07896-t003:** Correlation analysis between factors in surface water.

Factors	Temperature	Salinity	SS	pH	DO	AP	Cu	Pb	Cd	Zn	Hg	Cr	As
Temperature	1												
Salinity	−0.045	1											
SS	0.063	−0.232	1										
pH	0.344	−0.004	0.270	1									
DO	−0.350	0.355	−0.167	0.020	1								
AP	0.564 *	−0.396	0.124	0.337	−0.636 **	1							
Cu	0.032	−0.112	0.087	0.279	−0.449	0.541 *	1						
Pb	−0.539 *	−0.389	0.135	−0.056	−0.091	−0.058	0.295	1					
Cd	0.582 *	−0.205	0.441	0.192	0.070	0.175	−0.280	−0.566 *	1				
Zn	0.487	0.095	−0.068	−0.066	0.057	−0.020	−0.217	−0.141	0.284	1			
Hg	−0.100	−0.111	0.128	−0.231	−0.127	−0.277	0.185	0.207	−0.091	0.033	1		
Cr	−0.117	0.406	−0.242	0.467	0.102	0.122	0.214	−0.114	−0.347	−0.182	−0.258	1	
As	-0.248	0.043	−0.024	−0.506 *	0.064	−0.407	−0.563 *	−0.171	−0.033	−0.016	0.276	0.077	1

Note: * At the 0.05 level (two-tailed), the correlation is significant; ** At the 0.01 level (two-tailed), the correlation is significant. Number of valid cases *n* = 16. AP refers to active phosphate.

**Table 4 ijerph-19-07896-t004:** Correlation analysis between factors in bottom water.

Factors	Temperature	Salinity	SS	pH	DO	AP	Cu	Pb	Cd	Zn	Hg	Cr	As
Temperature	1												
Salinity	−0.882 **	1											
SS	0.191	−0.225	1										
pH	0.766 **	−0.826 **	0.223	1									
DO	0.103	0.068	−0.059	0.262	1								
AP	−0.249	0.183	−0.263	−0.414	−0.734 **	1							
Cu	0.187	−0.035	−0.463	0.143	0.059	0.294	1						
Pb	0.566 *	−0.250	−0.072	0.490	0.700 **	−0.570 *	0.177	1					
Cd	−0.280	0.187	−0.138	−0.273	−0.404	0.369	0.014	−0.319	1				
Zn	−0.068	−0.149	0.060	0.205	−0.285	0.068	0.222	−0.333	0.117	1			
Hg	0.157	0.000	−0.179	0.216	0.705 **	−0.378	0.171	0.623 **	−0.487	0.083	1		
Cr	0.225	−0.022	−0.281	0.026	−0.023	0.304	0.643 **	0.249	−0.213	−0.005	0.262	1	
As	−0.274	0.376	0.088	0.088	0.500*	−0.316	−0.129	0.381	−0.116	−0.128	0.417	0.046	1

Note: * At the 0.05 level (two-tailed), the correlation is significant; ** At the 0.01 level (two-tailed), the correlation is significant. Number of valid cases *n* = 16. AP refers to active phosphate.

**Table 5 ijerph-19-07896-t005:** Correlation analysis between factors in surface sediments.

Factors	OC	Sulfide	TPH	Cu	Pb	Cd	Zn	Hg	Cr	As
OC	1									
Sulfide	0.903 **	1								
TPH	0.242	0.246	1							
Cu	0.858*	0.886 **	0.081	1						
Pb	0.562	0.613	0.726	0.569	1					
Cd	0.583	0.772 *	0.340	0.687	0.373	1				
Zn	0.887 **	0.923 **	0.159	0.913 **	0.682	0.572	1			
Hg	0.850 *	0.973 **	0.224	0.925 **	0.626	0.813 *	0.940 **	1		
Cr	0.928 **	0.952 **	0.280	0.923 **	0.606	0.793 *	0.927 **	0.968 **	1	
As	0.312	0.380	0.550	0.572	0.747	0.446	0.404	0.462	0.436	1

Note: * At the 0.05 level (two-tailed), the correlation is significant; ** At the 0.01 level (two-tailed), the correlation is significant. Number of valid cases *n* = 7. TPH stands for Petroleum Hydrocarbons.

**Table 6 ijerph-19-07896-t006:** Heavy metal pollution index in surface sediments.

Sampling Station	$C_{f}^{i}$ **Pollution Index for the Signal Heavy Metal**							Comprehensive Pollution Index
	Cu	Pb	Cd	Zn	Hg	Cr	As	$C_{d}$
1	0.51	0.30	0.12	0.30	0.07	0.42	0.44	2.15
2	0.39	0.26	0.10	0.27	0.05	0.37	0.37	1.80
3	0.43	0.49	0.10	0.35	0.09	0.52	0.47	2.44
8	0.17	0.32	0.14	0.23	0.04	0.19	0.40	1.48
9	0.47	0.31	0.14	0.38	0.12	0.58	0.29	2.30
11	0.95	0.45	0.64	0.46	0.22	0.97	0.51	4.20
15	0.77	0.44	0.10	0.43	0.14	0.60	0.48	2.95
Ave	0.53	0.37	0.19	0.35	0.10	0.52	0.42	2.48

**Table 7 ijerph-19-07896-t007:** The potential ecological hazard index of heavy metal pollution in surface sediments.

Element		$E_{r}^{i}$	
	Max	Min	Ave
Cu	4.729	0.829	2.629
Pb	2.433	1.317	1.835
Cd	19.2	3	5.7
Zn	0.462	0.229	0.346
Hg	8.8	1.6	4.1
Cr	1.943	0.375	1.041
As	5.065	2.94	4.23
RI	42.5	12.8	19.9

**Table 8 ijerph-19-07896-t008:** Evaluation criteria for potential ecological risk coefficients of heavy metals in surface sediments.

$E_{r}^{i}$	Single Factor Potential Ecological Risk Classification	RI	Comprehensive Potential Ecological Risk Classification
$E_{r}^{i}<40$	Low	RI < 140	Low
$40\;\leq \;E_{r}^{i}\;$< 80	middle	140 ≤ RI < 280	middle
$80\;\leq \;E_{r}^{i}\;$< 160	higher	280 ≤ RI < 560	higher
$E_{r}^{i\;}\;$≥ 160	very high	RI ≥ 560	Very high

**Table 9 ijerph-19-07896-t009:** Contents of heavy metals in fishes (*n* = 6) obtained from Haikou Bay and adjacent waters Unit: mg/kg.

Station	Type of Fish	Cu	Pb	Zn	Cd	Cr	Hg	As
3	Setipnnataty	<1.0	0.083	7.9	0.017	<0.02	0.084	0.58
8	Narcine maculata	<1.0	0.21	6.2	0.0078	0.033	0.043	0.13
9	Johnius belengerii	<1.0	0.076	7.9	0.0093	<0.02	0.082	0.25
9	Taiustumifrons	<1.0	<0.005	2.3	0.00027	<0.02	0.095	0.36
11	Decapterus lajang	<1.0	0.04	6.4	0.0033	0.04	0.055	0.16
15	Setipnnataty	2.1	0.037	12	0.0049	<0.02	0.026	0.26
	Average	-	-	7.12	0.0071	-	0.064	0.29
Legislative threshold		20	2	40	0.6	5	1.5	0.3

Note: Legislative threshold refers to the quality standard value of heavy metals in marine organisms in China.

**Table 10 ijerph-19-07896-t010:** Target hazard quotient (THQ) and Hazard index (HI) of heavy metals in fishes (*n* = 6) obtained from Haikou Bay and adjacent waters.

Station	Type of Fish	Cu	Pb	Zn	Cd	Cr	Hg	As	HI
3	Setipnnataty	-	0.008	0.0107	0.0069	-	0.068	0.783	0.877
8	Narcine maculata	-	0.021	0.0084	0.0032	0.0045	0.035	0.176	0.248
9	Johnius belengerii	-	0.008	0.0107	0.0038	-	0.066	0.338	0.427
9	Taiustumifrons	-	-	0.0031	0.0001	-	0.077	0.486	0.566
11	Decapterus lajang	-	0.004	0.0086	0.0013	0.0054	0.045	0.216	0.280
15	Setipnnataty	0.021	0.004	0.0162	0.0020	-	0.021	0.351	0.415

**Table 11 ijerph-19-07896-t011:** Carcinogenic risks (CR) of heavy metals in fishes (*n* = 6) obtained from Haikou Bay and adjacent waters Unit: mg/kg.

Station	Type of Fish	Pb	Cd	As
3	Setipnnataty	8.8 × 10^−5^	8.1 × 10^−4^	1.1 × 10^−3^
8	Narcine maculata	2.2 × 10^−4^	3.7 × 10^−4^	2.4 × 10^−4^
9	Johnius belengerii	8.1 × 10^−5^	4.4 × 10^−4^	4.7 × 10^−4^
9	Taiustumifrons	-	1.3 × 10^−5^	6.7 × 10^−4^
11	Decapterus lajang	4.2 × 10^−5^	1.6 × 10^−4^	3.0 × 10^−4^
15	Setipnnataty	3.9 × 10^−5^	2.3 × 10^−4^	4.9 × 10^−4^

## Data Availability

Not applicable.

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
