# Peer review of "Characteristics and Health Risk Assessment of Heavy Metal Pollution in Haikou Bay and Adjacent Seas"

_ijerph, 2022, doi:10.3390/ijerph19137896_

Round 1

Reviewer 1 Report

The paper analyzed the characteristics and health risk assessment of heavy metal pollution in Haikou Bay and adjacent seas. The study is important to understand the pollution status in this region. It is recommended for publication after a major revision.

There are no complete page and line numbers in the manuscript to comment on more easily.

1. The abstract is confusing in the writing. For example,"In general, the results showed that...." sounds like the starting sentence to describe all the results in the abstract. But it is located right after the first results of "The concentration of heavy metals in the southern part of Haikou...".  There are too many "In addition, Furthermore, However, also..." which makes the abstract hard to understand. Please re-organize the abstract to make it more logic.

2. "seawater"  should be included in the "Keywords".

3. The last paragraph in the intro should be incorporated into the previous paragrpah.

4. there is no analysis of T, Sal, OC, Sulfide, DO, AP..... in the method section. 

5. Section 2.3 is too confusing to read. Please shorten, classify and pick the most related parameter for the writing. 

5. L163-164, the conclusion based on one point might not be solid. Please revise.

6 All fonts of figures are too small to read.

7. check  Figure 2, a ? b?

8. Contour map of Figure 3 based on 7 sites could be problematic. Please change the figure type. 

Author Response

Response to Reviewer 1 Comments

We would like to thank the editor sincerely for giving us a chance to resubmit the paper and also greatly appreciate the reviewers’ suggestions which would help us to improve the quality of the paper. We have made revision on our manuscript according to the editor's and the reviewers’ kind advice. The following is a point-to-point response to the editor’s and the reviewers’ comments and suggestions. We have added page and line numbers to the revised manuscript, and we apologize for our earlier failure to add them.

Point 1: The abstract is confusing in the writing. For example,"In general, the results showed that...." sounds like the starting sentence to describe all the results in the abstract. But it is located right after the first results of "The concentration of heavy metals in the southern part of Haikou...".  There are too many "In addition, Furthermore, However, also..." which makes the abstract hard to understand. Please re-organize the abstract to make it more logic.

Response 1: We apologize for the poor language of our manuscript. We worked on the manuscript for a long time and the repeated addition and removal of sentences and sections obviously led to poor readability. We have now worked on both language and readability and have also involved native English speakers for language corrections. We really hope that the flow and language level have been substantially improved. For the questions raised by the reviewers, we have made corresponding changes. For example, L8-11. “For seawater, the results showed that ……, port shipping, and ocean current movement.” Here we further specify the applicable object of the conclusion. We reorganized the abstract to make the sentences more logical.

Point 2:  "seawater"  should be included in the "Keywords".

Response 2: We agree with this suggestion and added it to the keyword.

Point 3: The last paragraph in the intro should be incorporated into the previous paragrpah.

Response 3: Thanks for the reviewer’s suggestion. We have incorporated this into the previous paragraph in the revised manuscript.

Point 4: there is no analysis of T, Sal, OC, Sulfide, DO, AP..... in the method section.

Response 4: We thank the reviewer for pointing out this issue. We indeed should add an analysis of these parameters in the method section. L126-130, L138-142: The contents of sulfide and petroleum hydrocarbon (TPH) in sediment were measured by spectrophotometry.

Point 5: Section 2.3 is too confusing to read. Please shorten, classify and pick the most related parameter for the writing.

Response 5: Section 2.3 have been corrected. We will be happy to edit the text further, based on helpful comments from the reviewers. On reflection, we believe that these parameters are indispensable. So we have made these parameters in a table to make it clearer. L214-

Point 6: All fonts of figures are too small to read.

Response 6: We thank the reviewer for pointing out this issue. We have redrawn all the figures to make the font easier to read. Fig. 1. – Fig. 4.

 Point 7: check  Figure 2, a ? b?

Response 7: We thank the reviewer for pointing out this issue. We have checked and made corresponding modifications and adjustments in the revised manuscript. Fig. 2. Spatial distribution of concentrations of seven trace metals in surface and bottom water bodies (a is the surface layer; b is the bottom layer).

 Point 8: Contour map of Figure 3 based on 7 sites could be problematic. Please change the figure type. 

Response 8: We thank the reviewer for pointing out this issue. We have changed the type of Figure 3 in the revised manuscript.

Reviewer 2 Report

The authors presented results of the study concerning the content of 7 metals in water (surface, bottom), sediments, and fish. The topic is relevant and the presented results may be of interest to readers. To improve the quality of the presented material, it is necessary to make a number of corrections.

1. In the section Materials and methods, it is necessary to add a description of the methods used to determine salinity, DO, SS, AP; How many fish samples of each species were analyzed?

2. It is necessary to provide in the tables the data that were used in the calculation of indices according to formulas 1-5, give the units of measurement of the indicators included in the formula

3. Why is LT much higher than ?? and ??? The difference is several orders of magnitude. Are the legislation thresholds (LT) set for surface or bottom waters?

4. Fig. 2-3 axes and legends are hard to read?

5. Table 4. What is PHC? Decryption required

6. Table 5-6: provide all the data for calculating the indices

7. Table 8. How many fish samples of each species were analyzed? Need to give SD

8. Table 9. It is necessary to give the sum of THQ. Why was the carcinogenic risk for Pb, Cd, As not calculated?

9. Has a correlation analysis been carried out between the content of metals in fish and the content of metals in water and sediment?

Author Response

Response to Reviewer 2 Comments

We would like to thank the editor sincerely for giving us a chance to resubmit the paper and also greatly appreciate the reviewers’ suggestions which would help us to improve the quality of the paper. We have made revision on our manuscript according to the editor's and the reviewers’ kind advice. The following is a point-to-point response to the editor’s and the reviewers’ comments and suggestions.

Point 1: In the section Materials and methods, it is necessary to add a description of the methods used to determine salinity, DO, SS, AP; How many fish samples of each species were analyzed?

Response 1: We thank the reviewer for pointing out this issue. We have added this part in the revised manuscript. For example, L126-130, L138-142: The contents of sulfide and petroleum hydrocarbon (TPH) in sediment were measured by spectrophotometry. We analyzed one sample for each type of fish, for a total of 6 samples from 5 species. We have clearly marked the number of samples in the table in the revised manuscript.

Point 2:  It is necessary to provide in the tables the data that were used in the calculation of indices according to formulas 1-5, give the units of measurement of the indicators included in the formula.

Response 2: We thank the reviewer for pointing out this issue. We will be happy to edit the text further, based on helpful comments from the reviewers. These parameters are put in a table after revision for clarity. And we provide the corresponding data in the revised attachments. The units of measurement are also marked in their respective tables and calculation sections. For excample, Table 1.

Point 3: Why is LT much higher than ?? and ??? The difference is several orders of magnitude. Are the legislation thresholds (LT) set for surface or bottom waters?

Response 3: Thanks for the reviewer’s suggestion. Legislative thresholds (LT) are regulations for heavy metal content in seawater. The comprehensive evaluation method is the result of formula conversion on the basis of the legislative standard value, and the result is used to determine the seawater quality. LT apply for both surface and bottom seawater.

Point 4: Fig. 2-3 axes and legends are hard to read?

Response 4: We thank the reviewer for pointing out this issue. We have redrawn all the figures to make the font easier to read. Fig. 1. – Fig. 4.

Point 5: Table 4. What is PHC? Decryption required

Response 5: We thank the reviewer for pointing out this issue. We have checked and made corresponding modifications and adjustments in the revised manuscript. A more suitable expression for PHC is TPH, we have made a note at the end of Table 5 ( previous Table 4). Such as, P 18/25.

Point 6: Table 5-6: provide all the data for calculating the indices

Response 6: We thank the reviewer for pointing out this issue. And we will provide the corresponding data in the revised attachments.

Point 7: Table 8. How many fish samples of each species were analyzed? Need to give SD

Response 7: We thank the reviewer for pointing out this issue. We analyzed a total of 6 samples from 5 species of fish. We have annotated in the corresponding table. For example, Table 9, Contents of heavy metals in fishes (n=6) obtained from Haikou Bay and adjacent waters.

Point 8: Table 9. It is necessary to give the sum of THQ. Why was the carcinogenic risk for Pb, Cd, As not calculated?

Response 8: We thank the reviewer for pointing out this issue. We apologize for ignoring this in the previous manuscript. We have added the sum of THQ in the revised manuscript and calculated the carcinogenic risk of Pb, Cd and As.

Point 9: Has a correlation analysis been carried out between the content of metals in fish and the content of metals in water and sediment?

Response 9: Thank you reviewer for pointing this out. To be honest, the correlation analysis between the metal content in fish and the metal content in water and sediment was also considered in the early stage of the study, our fish samples were very limited (only 6). And due to space considerations, we did not conduct a correlation analysis in the end. In future studies, we will consider analyzing the correlation of these three when the number of samples is appropriate. Thanks for your valuable advice.

Reviewer 3 Report

Very interesting article. I can see the tremendous amount of work and development of the publication. What is missing is guidance on what should be done to reduce this contamination, wha will be excellent summary of this hard work.

Author Response

Response to Reviewer 3 Comments

We would like to thank the editor sincerely for giving us a chance to resubmit the paper and also greatly appreciate the reviewers’ suggestions which would help us to improve the quality of the paper. We have made revision on our manuscript according to the editor's and the reviewers’ kind advice. The following is a point-to-point response to the editor’s and the reviewers’ comments and suggestions.

Point 1: Very interesting article. I can see the tremendous amount of work and development of the publication. What is missing is guidance on what should be done to reduce this contamination, wha will be excellent summary of this hard work

Response 1: We thank the reviewer for pointing out this issue. We will be happy to edit the text further, based on helpful comments from the reviewers. We have added an indicator of carcinogenic risk (CR) and a recommendation for consumers in the conclusion section to give guidance on how to reduce the impact of heavy metals on consumers. Thanks again to the reviewers for acknowledging our work.

Round 2

Reviewer 2 Report

The authors answered all the questions and made adjustments in accordance with the comments